# The Impact of Standard Care Versus Intrinsic Relaxation at Home on Physiological Parameters in Patients with Fibromyalgia: A Comparative Cohort Study from Romania

**DOI:** 10.3390/medicina61020285

**Published:** 2025-02-07

**Authors:** Theodora Florica Borze (Ursu), Annamaria Pallag, Emilian Tarcău, Doriana Ioana Ciobanu, Felicia Liana Andronie-Cioară, Carmen Delia Nistor-Cseppento, Gabriela Ciavoi, Mariana Mureșan

**Affiliations:** 1Doctoral School of Biomedical Sciences, Faculty of Medicine and Pharmacy, University of Oradea, 410073 Oradea, Romania; borze.theodoraflorica@student.uoradea.ro (T.F.B.); gciavoi@uoradea.ro (G.C.); mmuresan@uoradea.ro (M.M.); 2Department of Physical Education, Sport and Physical Therapy, Faculty of Geography, Tourism and Sports, University of Oradea, 410087 Oradea, Romania; emilian.tarcau@didactic.uoradea.ro (E.T.); dciobanu@uoradea.ro (D.I.C.); 3Department of Pharmacy, Faculty of Medicine and Pharmacy University of Oradea, 410073 Oradea, Romania; 4Department of Psycho-Neuroscience and Recovery, Faculty of Medicine and Pharmacy University of Oradea, 410073 Oradea, Romania; fcioara@uoradea.ro (F.L.A.-C.); dcseppento@uoradea.ro (C.D.N.-C.); 5Department of Dental Medicine, Faculty of Medicine and Pharmacy University of Oradea, 410073 Oradea, Romania; 6Department of Preclinical Disciplines, Faculty of Medicine and Pharmacy University of Oradea, 410073 Oradea, Romania

**Keywords:** fibromyalgia, sleep quality, fatigue, chronic pain

## Abstract

*Background and Objectives*: Fibromyalgia (FM), through the presence of widespread chronic pain, stiffens the musculoskeletal system and causes sleep disturbances and fatigue. Through this study, we aimed to compare the effectiveness of two different recovery interventions for improving sleep quality: a standard, multidisciplinary intervention in a recovery hospital versus a therapy focused on intrinsic relaxation at home. *Materials and Methods*: This study included 60 adult patients who participated voluntarily and were diagnosed with FM by a rheumatologist, randomly divided into two groups. During this study, 30 patients out of the 60 were randomly assigned to experimental group 1 and underwent treatment at the Recovery Clinical Hospital in Băile Felix. The other 30 patients were assigned to experimental group 2 and underwent treatment at home. They were assessed on the first and last day of the recovery program using the Fatigue Severity Scale (FSS) and the Pittsburgh Sleep Quality Index (PSQI). *Results*: In experimental group 1, where by patients underwent hospital recovery (EG1), the results show that the severity of fatigue (FSS) was significantly reduced, with *p* = 0.00 and an effect size of 0.77, which suggests a general improvement in the state of fatigue, as well as in the quality of sleep evaluated with the PSQI (*p* = 0.00, effect size = 0.55). In experimental group 2 (EG2), no change was observed between assessments in terms of the FSS, but in terms of the quality of sleep, there was a small decrease in the PSQI score (*p* = 0.083), with a small effect size of 0.09. *Conclusions*: The results show that, from a clinical point of view, a complex treatment carried out daily improves sleep quality and reduces fatigue.

## 1. Introduction

Fibromyalgia is a complex syndrome characterized by chronic pain, joint stiffness, fatigue, sleep disturbances, brain dysfunction, and depression [1,2]. The history of fibromyalgia is complex and has evolved gradually as doctors and researchers have tried to better understand the condition. However, the terms and explanations varied a lot [3]. In 1970, FM was associated with a central nervous system condition [4]; then, in 1976, the link between sleep disorders and chronic pain was first described [5]. In 1981, the Yunus criteria included in FM diagnosis comprised fatigue, sleep problems, sensitivity to weather, and chronic pain, all of which are aggravated by the presence of stress and anxiety [6]. Patients with fibromyalgia have been observed to have an increased sensitivity to stimuli that would not normally cause pain, a phenomenon known as allodynia [7]. In addition, research has shown that the neurotransmitters responsible for pain, such as substance P and serotonin, are involved in the exacerbation of symptoms [8]. Reflex phenomena are considered among the most important factors triggering FM, and they are associated with deep chronic pain produced by the activation of a pain–spasm cycle through the presence of repeated trauma or postural stress [9]. The exact causes and pathogenic mechanisms of fibromyalgia are unknown. Biological, psychological, and socio-cultural factors have been suggested as possible contributors, including abnormal pain signaling, abnormal activity of the neuroendocrine and autonomic systems, genetic predisposition, certain environmental triggers, and sleep disturbances [10]. These factors hypersensitize patients to pain [11]. Since the pathogenesis of this disease is not well known, the diagnosis is only made clinically at this time. Oxidative stress, mitochondrial dysfunction, mineral and vitamin deficiencies, and disproportions in other components are interesting and attractive topics from a clinical point of view, which require further studies to clarify the status and development of FM [12]. It is necessary to find the most appropriate solutions to reduce the permanent fatigue faced by these patients.

The European League against Rheumatism states that, for the management of FM, physical exercises represent the only effective recommendation over time, being adapted according to the needs of patients, with some patients not being able to tolerate strength and flexibility exercises even though they seem to be the most beneficial [13]. Regarding the management of FM [14], treatment requires a comprehensive approach that combines non-pharmacological interventions, such as exercise, and pharmacological management with duloxetine, pregabalin, milnacipran, and amitriptyline, which have remarkable benefits for fibromyalgia symptoms. Pure mu-opioid agonists such as NSAIDs and acetaminophen do not have effective results, and methylphenidate, despite improving mental status and fatigue, requires further studies [15]. However, pharmacological therapy is only useful for individual symptoms [16], with non-pharmacological interventions being more effective according to systematic reviews, because combining aerobic exercise with resistance and stretching reduces FM symptoms and the stress response [17]. Some authors believe that accepting pain without daily attempts to reduce it has positive effects on improving daily functioning [18].

Psychological factors should be considered alongside the management of FM, as they contribute significantly to both symptomatology and the functioning of the treatment. Relaxation therapy is an autonomous process, representing a main goal in the prophylaxis and recovery of a patient, because it brings a state of rest to the muscles and inhibits nervous tension. Intrinsic relaxation rebalances the well-being inside the body. Studies on mind–body recovery are limited, and further research is needed. Published studies have limited effects due to low-quality evidence or lack of recent studies [19]. At the same time, mind–body therapies are favorable in pathologies that have affected psychological function [20], aiming to induce the body’s relaxation response [21]. The author Liraz Cohen-Biton concludes that holistic therapies focusing on body, mind, and soul provide an appropriate response that promotes the health status of patients with FM [22]. A study of this therapy was conducted with 28 women with FM via the Zoom platform. This study showed that performing eight sessions of conscious breathing and listening to one’s own body for 8 weeks reduced physical and mental suffering, fatigue, fear, and disabilities, *p* < 0.001 [23]. During home therapy, FM patients were assessed on several dimensions such as the presence of stress in their daily environment, program resilience, and coping to check the effectiveness of treatments and to understand the weaknesses they face [24]. Also, according to Theadom [25], mind–body therapies could be good therapeutic strategies for physical and psychological improvement because mind, body, and behaviors are interconnected. Another purpose of using relaxation through online consultations is the permanent adherence of patients to recovery, something that would be impossible during pandemics such as SARS-CoV-2, which negatively affected the management of chronic health conditions by imposing isolation restrictions [26,27,28].

Regarding sleep, it is known that there are two types of sleep, i.e., REM and non-REM. The non-REM period has four stages with waves that have certain electroencephalographic characteristics [29,30]. In fibromyalgia, there are disturbances in the fourth phase of non-REM sleep, which are translated by the appearance of alpha waves that overlap the normal delta waves. The results from specialized literature show that sleep disorders are one of the most important aspects of FM [31] poor sleep quality being a fundamental index in the pathophysiology of FM, together with fatigue and pain. The prevalence of sleep quality shows that only 11% of FM patients reported good sleep quality, while up to 99% of patients report having problems with the amount, initiation, and maintenance of sleep during the night, with long refresh times or waking up too early in the morning, which makes it difficult to start their daily activities [32]. Kline et al. [33] mentioned that the definition of sleep quality should be related to the personal experience of each person, depending on the stage that affects them. Poor sleep quality has a negative impact because it lowers the pain threshold and increases the level of its catastrophizing. At the same time, this cycle of sleep impairment causes a high level of fatigue; studies show that poor sleep quality leads to the appearance of cognitive, social, and integration problems [34], having a tendency to become chronic [35]. Patients with fibromyalgia complain of sleep disturbances and the intensification of pain, stiffness, and fatigue after waking up [36]. Often, when patients are asked if they wake up rested, the answer is negative, with sleep being impaired in over 80% of FM patients [37]. Often sleepless nights are followed by worsening pains [38]. Some studies suggest that fatigue is a cause of FM and not a symptom [39]. In the anamnesis, there are frequent associations with diseases that recognize a stress etiology, such as ulcerative hemorrhagic colitis, irritable colon, migraine, hypertension, and so on [40].

We created this study by observing the negative impact of FM on patients’ lives and the lack of evidence for the introduction of programs focused on relaxation. The aim of this study is to compare the effects of applying two different types of intervention on sleep quality and fatigue in fibromyalgia patients, respectively, a therapy without additional costs, focused only on intrinsic relaxation carried out at home in relation to a multidisciplinary therapy.

## 2. Materials and Methods

Following visits to the rheumatologist, 67 patients with fibromyalgia were invited, by the physical therapist, to participate in the present study. Seven of them withdrew before the start of the initial assessments at the time of the study presentation. A total of 60 patients participated in this study. Patients were chosen following the inclusion and exclusion criteria. The main inclusion criteria were the age of at least 18 years, the presence of a diagnosis of fibromyalgia, voluntary and independent participation in this study, and the absence of contraindications for the intervention. At the same time, pregnant women and those who were absent from the evaluation were excluded from this study.

The allocation of patients was carried out by the method of simple randomization, without knowing the severity of the symptoms, so that at the end of this study, the chance of specific recovery would be given to the group with more effective intervention. Furthermore, 30 patients out of 60 were randomly assigned to experimental group 1 and underwent treatment at the Recovery Clinical Hospital in Băile Felix. The other 30 patients were assigned to experimental group 2 and performed the treatment at home. The period of this study is January–November 2024. All 60 patients also administered the pharmacological treatment received during the study period upon learning the diagnosis.

The two groups received treatment for 2 weeks. EG1 performed 10 standard recovery sessions in the presence of a physical therapist. They had a complex recovery program consisting of 30 min of individual physical therapy that included warm-up exercises and mobilization exercises for all joints, followed by muscle-strengthening exercises and stretching at the end for increased flexibility [41,42,43,44,45,46,47]. Each exercise was performed in 4 sets of 10 repetitions. The exercises were performed from dorsal decubitus, ventral decubitus, standing, and quadrupedal positions and were performed in breathing rhythm. Patients also had electrotherapy sessions at the BTL 5000 machine with low frequency (80–130 Hz) transcutaneous electrical nerve stimulation (TENS) for 15 min to reduce chronic pain and classic Swedish massage for 10 min. They also had group hydrokinetotherapy for 20 min from orthostatism at a water temperature of 36–37 degrees Celsius, which facilitates movement and reduces gravitational forces [48], the benefits of thermal water being known since ancient times [49]. Exercise is part of managing fibromyalgia as it reduces pain and improves sleep and overall functioning [45]. Physiotherapy is the most important basis because it is based on movement, which is essential for life. Mobilizations, tractions, active exercises, active exercises with resistance, isometrics, and stretching are performed because people with FM have low endurance and muscle strength [46]. Massage brings a general and local state of well-being, reduces pain, and improves circulation and is useful when the patient’s mobility is reduced, helping to rebalance the posture, improve muscle relaxation, and eliminate acute pain [47]. Electrotherapy reduces pain and can be used for medical purposes because the human body works mostly electrically [50]. Hydrokinetic therapy is one of the main forms of restoring diminished functions because water can support 90% of the body’s weight, reducing the muscle effort required to perform movements and increasing the amplitude of movements [48].

Experimental group 2 (EG2) performed at home for 2 weeks without the physical supervision of a physiotherapist, 20 min of intrinsic relaxation specific to the author Parow, every evening before going to bed. Parow’s relaxation technique is based on intrinsic relaxation, which helps to actively induce relaxation and ensures mutual inhibition between muscles and psyche. It is performed without other auxiliary tools and can be performed at home. The author recommends maintaining the supine position in bed for 20 min, breathing freely. During this time, it is recommended to inhale through the nose and exhale with a long “sh” to automatically induce general muscle relaxation. All patients in this group were helped before the start of the 10 recovery sessions to perform the breathing technique correctly and communicated daily, online, with the physiotherapist to record the session performed and to offer them help if they needed it. This intervention, focused only on intrinsic relaxation, is based on the need to integrate a recuperative program as simplified as possible and accessible to anyone, the specialists in the field claiming that notable improvements in the clinical profile occur in people who have a high adherence to the treatment.

Each group was analyzed after signing consent for the processing of personal data to ensure voluntary participation. After signing the documents, the first assessment was carried out one day before the start of the treatment, and the final assessment was carried out at the end of the recovery sessions, after the two weeks of treatment.

The data collected were used for this study only, and patients could withdraw from this study at any time. In the studies, only patients’ initials were used, and no pictures were taken during this period. To respect their right to privacy, the recovery was carried out only in the presence of the physical therapist.

### 2.1. Instruments

To find out information about the subjects, an assessment form was drawn up that included questions about independent variables such as age, height, sex, weight, duration of the condition, associated diagnoses, symptoms, triggers, background, occupation, and medication. The following two questionnaires were assigned to the dependent variables: the Fatigue Severity Scale (FSS) questionnaire to assess fatigue and the Pittsburgh Sleep Quality Index (PSQI) for sleep quality.

#### 2.1.1. Assessment of Fatigability and Fatigue

Fatigue severity was assessed using the Fatigue Severity Scale (FSS) [51]. The Fatigue Scale is used to measure the severity, frequency, and impact of fatigue, typically in patients with chronic conditions such as cancer, multiple sclerosis (MS), or other long-term illnesses such as fibromyalgia. Fatigue is a common symptom that affects a person’s physical, emotional, and mental well-being, and these scales help clinicians and researchers provide better diagnosis, monitoring, and treatment. The scale is self-reported, requiring only paper and pen. Subjects responded to the 9 items using a 7-point Likert scale (1 = strongly disagree, 7 = strongly agree). The minimum score is 9 and the maximum is 63, which indicates a severe state of fatigue.

#### 2.1.2. Assessment of Overall Sleep Quality

Sleep quality has a clinically relevant influence on a person’s daytime functioning. The degree of sleep disturbance has been studied most with the help of the Pittsburgh Sleep Quality Index (PSQI) questionnaire [52], which comprises 19 items related to 7 subcategories: subjective sleep quality, latency period, sleep duration, usual sleep efficiency, sleep disorders, sleep medication use, and daytime dysfunction. Five additional roommate-rated questions are included for clinical purposes but are not scored. The questionnaire is validated for patients with major depressive disorders, sleep initiation and maintenance disorders, excessive sleepiness disorders, insomnia, depression, chronic pain, and other health conditions, cancer, and fibromyalgia. It is also used in research studies to investigate how sleep quality is related to various health outcomes. The questionnaire consists of a combination of Likert-type and open-ended questions that will later be converted to scaled scores using guidelines provided by the authors. Patients are asked to indicate how often they have experienced certain sleep difficulties in the last month. Scores for each question range from 0 to 3, with higher scores indicating more disturbed sleep [53]. The component scores are then summed to give a global PSQI score, ranging from 0 to 21, where scores ≤ 5 indicate good sleep quality, scores > 5 indicate poor sleep quality, and scores > 10 suggest a potential sleep disorder.

### 2.2. Statistical Analysis

The results were interpreted, and the statistical analysis aimed at improving the variables after treatment for each group. The IBM SPSS Statistics for Windows software, Version 29.0, was used for the statistical processing of the study data (30-day trial version; Armonk, NY, USA: IBM Corp). Nominal data were described as frequency and percentage and continuous variables as mean, median, standard deviation, frequency ranges, minimum, and maximum. Differences between groups and time points were assessed using Student’s *t*-test and Chi-square test, with a significance threshold of *p* < 0.05 and high significance defined as *p* < 0.01. The independent samples *t*-test was used to compare the means with respect to the dichotomous variables in this study. To compare two dependent or paired datasets, we used the *t*-test for dependent (paired) samples. The *t*-test for dependent samples (pairs) is used in research protocols that involve repeated measurements on the same individuals or on individuals with similar characteristics (even twins). The data are considered paired because for each value there is a matched value. The test evaluates the difference score within each pair so that subjects are compared only with themselves or their pair. The *t*-test of the difference between the means of two dependent samples allows the evaluation of the significance of the variation of a certain characteristic in the same individuals in two different situations (for example, “before” and “after” the action of a certain condition, or in two different contexts, regardless of the time of their manifestation). The major advantage of this statistical model is that it captures the so-called “intrasubject” variation by the fact that the basis of calculation is the difference between the two values of each individual subject.

### 2.3. Sample Size

The general formula for calculating the size of the patient sample used in this study was is expressed as follows:*n* = (Zα/2 + Zβ)2·(2σ2)/d2, 
where

*n* is the sample size.Zα/2 = 1.96 is the critical value of the standard normal distribution (Z) corresponding to the significance level α (0.05).Zβ = 0.80 is the critical value of the standard normal distribution corresponding to the power of the test (1 − β).σ is the population standard deviation.d is the minimum effect size we want to detect.

Thus, *n* = (1.96 + 0.84)2⋅(2·102)/52 = 62.7.

In this study, 67 patients were recruited.

## 3. Results

The results include the evaluation of the two analyzed groups.

### 3.1. Patient Data

In Table 1, the average age difference between EG1 (46.97 years) and EG2 (44.73 years) is 2.233 years, and the *p*-value is 0.484. This is a relatively small difference, suggesting that, in terms of age, the two groups are comparable.

The gender distribution, observed in Table 2, shows a female prevalence of FM in both EG1 and EG2. In other words, there is no significant association between gender and the group to which the participants are assigned.

Below, in Table 3, it can be seen that all patients followed a pharmacological treatment based on non-steroidal anti-inflammatory drugs, which shows the homogeneity of the study groups.

The distribution by body mass index (BMI) shows, in Table 4, that the *p*-value (0.426) is much higher than the usual significance threshold of 0.05, indicating that the difference between the groups is not statistically significant. Thus, there is insufficient evidence to state that the BMI between experimental groups 1 and 2 is significantly different.

Table 5 shows the distribution of symptoms reported by participants in the two groups. Both groups experience a significant number of symptoms, with a similar prevalence for many of them. However, EG1 reports a higher prevalence for certain symptoms, such as migraines, paresthesias, and anxiety, and EG2 reports a higher prevalence for chronic fatigue. In general, symptoms such as chronic pain, fatigue, and sleep disturbances are common in both groups.

Table 6 shows the distribution of frequencies for the triggering factors according to the type of lot. The contingency table shows how many participants in each group reported each trigger. There is a diversity of triggers reported in the two groups, with psychological stress being the main common factor and the most frequently mentioned. Factors such as infections and physical trauma are less relevant in this analysis, and the experimental group tends to have a higher prevalence of psychological stress and hormonal changes.

### 3.2. The Results of the Evaluation of the State of Fatigue Evaluated with the FSS, Respectively of the Quality of Sleep Evaluated with the PSQI

In Table 7 and Figure 1, the intervention applied to EG1 shows that sleep duration increased, with patients able to fall asleep faster and maintain sleep for a longer period. Diurnal dysfunction caused by lack of sleep was reduced, with patients reporting better overall sleep quality with a moderate effect size of 0.55 points. Fatigue Severity (FSS) was significantly reduced, *p* = 0.00, with a moderate effect size of 0.77 points, suggesting an overall improvement in fatigue status. Overall, the intervention had a significant positive effect on various aspects of sleep and daytime functioning.

In EG2, the intervention had limited effects. No significant changes were observed between baseline and final values in most sleep components, such as sleep quality, sleep duration, sleep efficiency, sleep medication use, and Fatigue Severity Scale. The global sleep quality score (PSQI) decreased slightly, with a low effect size of 0.09 points, and the fatigue state remained at the same value.

In Table 8, regarding the comparison of the two groups at baseline, significant differences can be observed between EG1 and EG2 for sleep latency, sleep efficiency, PSQI global score, and FSS score. EG1 shows poorer sleep quality and more sleep-related problems compared to EG2. However, no significant differences were observed for sleep quality, sleep disturbances, sleep medication use, and daytime dysfunction. The results of the final evaluation show significant differences between the EG1 and EG2 regarding sleep latency and daytime dysfunction. EG1 had higher latency and lower diurnal dysfunction compared to EG2. Regarding sleep quality, sleep duration, sleep efficiency, sleep disturbances, use of sleep medication, PSQI total score, or FSS total score, no significant differences were observed between the two groups (*p* > 0.05). During the sleep analysis, it can be seen that the patients in EG2 with more negative symptoms have significant improvements regarding latency and daytime dysfunctions compared to the less symptomatic patients in group 2.

## 4. Discussion

The specialized literature confirms the need for more in-depth studies regarding the management of fibromyalgia, especially regarding beneficial therapeutic measures to improve the inner state. Although this study has some limitations, especially the short period of the intervention, significant results can be observed in the case of the multidisciplinary intervention, as the sessions were conducted daily. This study has clinical importance due to the novelty of comparing these two types of intervention.

In terms of age and gender correlation with treatment efficacy, there is no clinical connection, with improvement occurring on average equally across all ages.

The homogeneity of the groups in terms of age, gender, muscle mass index, pharmacological treatment, and felt symptoms demonstrates that the changes occurring during the treatment are due only to the recovery intervention.

In terms of symptoms, FM patients show symptoms of fatigue, headaches, paresthesias, myalgias, irritable bowel syndrome, micturition disorders, cognitive and sleep disorders including depression and anxiety that lead to the inability to perform daily activities [54]. These symptoms are also present in the study carried out, in different percentages, each having a role in maintaining chronic pain that leads to a state of general fatigue and disturbed sleep quality.

The use of massage as a decontracting means applied externally to the body helps to reduce the state of muscle tension and stops the stagnation of non-oxygenated blood; patients have a more efficient sleep, managing to fall asleep faster and maintain sleep for a longer period. Electrotherapy applied by means of low-frequency currents has effects on the production of endorphins, reducing pain, the results of the intervention indicating a significant reduction in the use of sleep medications. Exercises for resistance, muscle strengthening, and flexibility help to stimulate circulation and better oxygenation, which reduces fatigue and diminishes the diurnal dysfunctions that most patients face when performing daily activities. Thus, by using these techniques alongside hydrokinetotherapy, which through the action of water reduces physical stress, the fatigue–restless sleep cycle is inhibited. The quality of sleep being improved, the accumulated fatigue is also reduced. Overall, the intervention in the case of EG1 had beneficial effects on the studied variables.

The induction of relaxation through the internal rebalancing of the body based on breathing and muscle rest has limited effects (*p* > 0.05) in the short term on the quality of sleep, slightly improving by eliminating the state of tension during the period of maintaining sleep and helping to reduce diurnal dysfunctions and disorders that occur during sleep, but the difference compared to the first group is visible at the level of fatigue, the patients not having improvements in this chapter.

Regarding within-group variations, at EG1, the decrease in mean score indicates an improvement in patient-reported fatigue between the two assessments, with a general trend observed and improvement in fatigue symptoms for most participants, not just those with score extremes. The standard deviation decreased from 10.811 to 8.763, suggesting a leveling of fatigue severity across participants. At the beginning, the variability of the scores was greater, but at the end, the differences between the participants became smaller, which may indicate a generalized and more even improvement. The decrease in values for each component and the total PSQI suggests an overall improvement in sleep quality and a reduction in sleep-related problems following the intervention.

Intra-group variations within EG2 show that there is no change in the mean score between the initial and final assessment, suggesting that the intervention did not influence the participants’ perceived level of fatigue. Both the mean and median constants indicate that the fatigue felt by the group remained uniform before and after. The standard deviation of 12.820 indicates moderate variability between participants, both at baseline and at follow-up.

Referring to the purpose of this study, we can conclude that a multidisciplinary therapy has meaningful effects on FM patients, while the recovery based only on relaxation does not focus on all the weak points of the patients; in this case, the hypothesis is not confirmed. The results of this study have clinical relevance and help specialists in the field to improve management.

Contextualizing the obtained results, the influence of different recovery procedures on fatigue and sleep quality can be seen below. Fibromyalgia has a low incidence; most studies in the specialized literature are conducted on a small sample. The author Andrade C. [55] et al. looked at the effects of water training in women with fibromyalgia. After the 16 weeks of intervention, the amount of VO2 increased in the experimental group (*p* = 0.04), the pressure pain threshold increased (*p* = 0.02), but the effects on sleep quality were insignificant. In this study, it was concluded that aquatic exercises improve cardiovascular function but have no significant effects on the other variables. However, aquatic exercise combined with individual physical therapy, manual therapy, and electrotherapy by EG1 produced a significant difference (*p* = 0.00) between ratings, suggesting a significant decrease in sleep severity. In another article by Hauser et al. [56], it was concluded that aerobic exercises combined with flexibility and muscle-strengthening exercises do not have significant effects on sleep quality (SMD = 0.01), while Steffens et al. [57] showed that aerobic exercise combined with hydrokinetotherapy has a moderate effect on improving sleep quality. Also, Kundakci et al. [58] found that strength training has significant effects of improving sleep quality with a moderate clinical effect (SMD = −0.74), and Busch et al. [59] reached the same conclusion about aerobic exercises. The author Cuenca-Martinez F. et al. [60] explained that relaxation exercises combined with aerobic and strength exercises bring about a real improvement in the quality of sleep that will lead to an improvement in the quality of life, compared to recovery programs isolated.

The author Nadal et al. [61], following the effects of manual therapy, concluded that there are no significant improvements regarding FSS and PSQI, the *p*-value being greater than 0.05, but this form of therapy decreases the perception of pain, thus concluding that interventions based on relaxation, both intrinsic and extrinsic through massage, do not change the state of fatigue.

### Strengths and Limitations of This Study

As, to our knowledge, this is the first study in Romania comparing two types of intervention on patients diagnosed with FM, this is also the strength of our study.

This study has limitations regarding the short duration and the small number of patients due to the incidence of FM because the diagnosis is very difficult to make, and patients find out the exact cause over several years. Short intervention periods are due to country legislation because Romanian legislation only offers 10 free recovery sessions for 2 weeks, once every 6 months. To carry out this study over a longer period, it would have been necessary to change the location, which involved giving up hydrokinetotherapy. EG2 had the same limitation in terms of recovery time because it was desired to follow the effectiveness of two different programs in the same number of sessions. Conducting recovery sessions at home has both positive effects in terms of the psychological state and stress being eliminated, as well as negative effects caused by the lack of permanent supervision. These limitations, together with the performance of a single procedure alone, attracted lower clinical improvements compared to group 1.

To observe the long-term results, the patients in experimental group 1 will be re-evaluated 6 months after the intervention, and the patients in experimental group 2 received the recommendation to continue the relaxation technique at home for 6 months, monitored online by a physiotherapist to avoid biases. The purpose of continuing the program is to observe if a treatment based on a single form of relaxation can improve the state of fatigue by acting on sleep, following this period to begin multidisciplinary treatment if no beneficial results appear on the variables evaluated.

## 5. Conclusions

Due to the unclear pathogenesis of fibromyalgia, the multidisciplinary approach focused on improving flexibility, extrinsic relaxation, and muscle tone has better effectiveness on sleep quality than therapy focused only on intrinsic relaxation. Although the intervention was of a short duration, the daily sessions made an increased contribution to the recovery. This study offers a new perspective on the integration of simple recovery programs in chronic pathologies. The clinical relevance of our findings helps specialists have better management of FM, since better quality of sleep ultimately improves the quality of life.

## Figures and Tables

**Figure 1 medicina-61-00285-f001:**
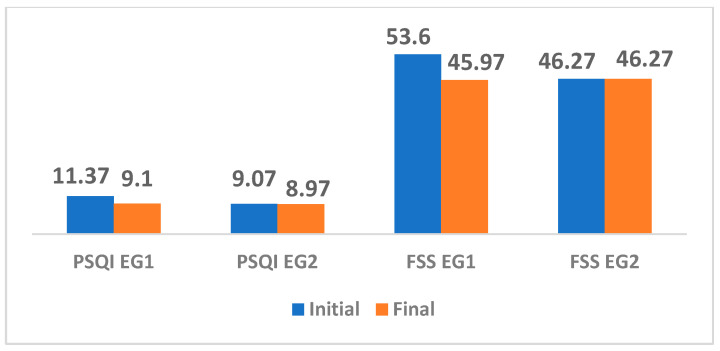
The results of FSS and PSQI scores from the two groups, EG1 and EG2.

**Table 1 medicina-61-00285-t001:** Distribution by age of patients.

Group	N	Mean	SD
EG1	30	46.97	13.163
EG2	30	44.73	11.317

SD = standard deviation.

**Table 2 medicina-61-00285-t002:** Distribution according to patient gender.

Group		Gender		Total
		Male	Female	
EG 1	Count	2	28	30
	%	6.7%	93.3%	100.0%
EG 2	Count	3	27	30
	%	10.0%	90.0%	100.0%
Total	Count	5	55	60
	%	8.33%	91.6%	100.0%

**Table 3 medicina-61-00285-t003:** Distribution according to pharmacological treatment.

Pharmacological Treatment	EG1 (Count/%)	EG2 (Count/%)	Total EG1 + EG2 (Count/%)
Ibuprofen	16 (52)	14 (48)	30 (50)
Acetaminophen	14 (48)	16 (52)	30 (50)

**Table 4 medicina-61-00285-t004:** Distribution of patients according to BMI.

Group	N	SD	Mean	t	df	*p*	Mean Difference
EG1	30	24.4207	4.41671	0.801	58	0.426	0.78600
EG2	30	23.6347	3.05794				

N = the number of patients; SD = standard deviation; t = the *t*-test of the difference in means; df = the difference between the two values; *p* = statistical significance coefficient.

**Table 5 medicina-61-00285-t005:** Distribution of symptoms reported by EG1 and EG2 participants.

Symptom ^a^	U.M.	Group		Total
		EG1	EG2	
Chronic diffuse pain	Count	29	29	58
	%	96.7%	96.7%	
Fatigability	Count	19	15	34
	%	63.3%	50.0%	
Headache	Count	15	13	28
	%	50.0%	43.3%	
Migraines	Count	12	7	19
	%	40.0%	23.3%	
Paresthesia	Count	14	11	25
	%	46.7%	36.7%	
Myalgia	Count	18	17	35
	%	60.0%	56.7%	
Irritable bowel syndrome	Count	7	6	13
	%	23.3%	20.0%	
Restless legs syndrome	Count	9	4	13
	%	30.0%	13.3%	
Cognitive disorders	Count	2	2	4
	%	6.7%	6.7%	
Sleep disorders	Count	14	15	29
	%	46.7%	50.0%	
Depression	Count	9	6	15
	%	30.0%	20.0%	
Anxiety	Count	17	12	29
	%	56.7%	40.0%	
Chronic fatigue	Count	19	23	42
	%	63.3%	76.7%	
Other	Count	2	0	2
		6.7%	0.0%	
Total	Count	30	30	60

^a^ Percentages and totals are based on respondents; U.M. = unit of measure.

**Table 6 medicina-61-00285-t006:** The distribution of patients according to the triggering factors.

The Triggering Factor		EG1	EG2	Total
I do not know	Count	7	9	16
	%	23.3%	30.0%	
Infections (parvovirus or Lyme)	Count	3	0	3
	%	10.0%	0.0%	
Physical trauma	Count	2	0	2
	%	6.7%	0.0%	
Psychological stress	Count	19	14	33
	%	63.3%	46.7%	
Hormonal changes	Count	8	5	13
	%	26.7%	16.7%	
Medicines and vaccines	Count	1	0	1
	%	3.3%	0.0%	
Catastrophic events (war, etc.)	Count	1	1	2
	%	3.3%	3.3%	
Other	Count	1	3	4
	%	3.3%	10.0%	

**Table 7 medicina-61-00285-t007:** Comparison of the parameters between the initial and final moment within EG1 and EG2, respectively.

Component	Paired-Sample Statistics						Paired-Sample Test		
			Mean		SD				
	Group	N	Initial	Final	Initial	Final	t	df	*p*
Sleep Quality	EG1	30	2.07	1.53	0.740	0.571	5.757	29	0.000
	EG2	30	1.83 ^a^	1.83 ^a^	0.834	0.834			
Sleep latency	EG1	30	1.67	1.37	1.155	0.928	3.525	29	0.001
	EG2	30	0.80	0.77	0.925	0.898	1.000	29	0.326
Sleeping duration	EG1	30	1.63	1.20	1.098	0.961	4.709	29	0.000
	EG2	30	1.20 ^a^	1.20 ^a^	0.761	0.761			
Sleep efficiency	EG1	30	1.73	1.10	1.143	0.995	4.289	29	0.000
	EG2	30	1.03 ^a^	1.03 ^a^	1.066	1.066			
Sleep disturbance	EG1	30	1.67	1.57	0.547	0.568	1.795	29	0.083
	EG2	30	1.57	1.53	0.504	0.507	1.000	29	0.326
Use of sleeping medication	EG1	30	0.63	0.47	0.890	0.776	2.408	29	0.023
	EG2	30	0.50 ^a^	0.50 ^a^	0.731	0.731			
Daytime dysfunction	EG1	30	1.97	1.53	0.765	0.571	4.709	29	0.000
	EG2	30	1.93	1.90	0.828	0.803	1.000	29	0.326

^a^ The correlation and t cannot be computed because the standard error of the difference is 0; N = the number of patients; SD = standard deviation; t = the *t*-test of the difference in means; df = the difference between the two values; *p* = statistical significance coefficient.

**Table 8 medicina-61-00285-t008:** Comparison of parameters between the two groups at baseline and endpoint (independent samples test).

Component	*p*	
	Initial	Final
Sleep Quality	0.256	0.109
Sleep latency	0.002	0.014
Sleeping duration	0.081	1.000
Sleep efficiency	0.017	0.803
Sleep disturbance	0.464	0.811
Use of Sleeping Medication	0.529	0.865
Daytime dysfunction	0.872	0.046
PSQI total	0.048	0.899
FSS total	0.020	0.916

*p* = statistical significance coefficient.

## Data Availability

Data is contained within the article.

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
