# Peer review of "The Impact of Standard Care Versus Intrinsic Relaxation at Home on Physiological Parameters in Patients with Fibromyalgia: A Comparative Cohort Study from Romania"

_medicina, 2025, doi:10.3390/medicina61020285_

Round 1

Reviewer 1 Report

Comments and Suggestions for Authors

Recommendations for Authors

Introduction

The introduction provides an adequate historical overview of fibromyalgia; however, it would benefit from a more structured presentation of the rationale behind the study.

Some references appear outdated; consider including recent studies to reflect current findings.

The introduction should more clearly outline the study's objectives and how they address gaps in the existing literature. 

Some sentences are lengthy and difficult to follow. In Lines 38-46, "Fibromyalgia is a complicated syndrome today due to the presence of characteristic symptoms: chronic pain, joint stiffness, fatigue, sleep problems, brain dysfunction and depression." The sentence structure can be simplified for readability. I suggested being revised to: "Fibromyalgia is a complex syndrome characterized by chronic pain, joint stiffness, fatigue, sleep disturbances, brain dysfunction, and depression."

Also, in Lines 58-67, there is an awkward phrasing: "It is considered that among the most important factors in triggering FM would be the reflex phenomena..." Suggested revision: "Reflex phenomena are considered among the most important factors triggering FM..."

Research Design

The study design is generally appropriate for the research objectives. However, there are concerns about the short intervention period (2 weeks), which may not be sufficient to observe meaningful changes.

The randomization process should be described in more detail to ensure transparency and reproducibility.

The inclusion of only two groups may limit the generalizability; consider discussing potential confounders in more depth.

There also some sentences are lengthy in Lines 165-174, "They had a complex recovery program consisting of 20-minute individual physical therapy that included aerobic exercise, stretching and muscle-strengthening exercises, 15-minute low-frequency electrotherapy, massage, and hydrokinesitherapy that facilitates movement." Please consider this revision, "Their recovery program included 20-minute sessions of physical therapy, comprising aerobic exercise, stretching, and muscle-strengthening exercises. Additionally, they received 15-minute sessions of low-frequency electrotherapy, massage, and hydrokinesitherapy."

Methods

The methods section is well-structured but lacks detail in the intervention protocol for intrinsic relaxation at home. Providing more specifics on adherence monitoring would be beneficial.

The statistical methods are generally appropriate; however, clarifications on how missing data were handled should be provided.

A power analysis to determine the sample size adequacy should be included.

Results

The results section provides statistical comparisons but lacks visualization (e.g., tables and figures) to support the textual descriptions.

Some p-values (e.g., p=0.083) are not significant, but their interpretation is unclear. Consider emphasizing clinically meaningful changes rather than purely statistical significance.

A discussion of within-group variations should be expanded.

In Lines 328-334, this sentence is complex: "In the experimental group 1 that performed hospital recovery (EG1), the results show that the severity of fatigue (FSS) was significantly reduced, which suggests a general improvement in the state of fatigue." Please consider revising it. 

Conclusions

The conclusions align with the study findings, but they should better acknowledge the limitations regarding the short intervention period and sample size.

Recommendations for future research should be expanded to include longer follow-up periods and additional outcome measures.

Overall, the manuscript requires language improvements, including grammatical corrections, simplification of complex sentences, and improved clarity.

Comments on the Quality of English Language

Some sentences are lengthy and difficult to follow. Revising for clarity and conciseness is recommended. Minor grammatical errors and awkward phrasing should be corrected to enhance readability.

Author Response

Dear Reviewer,

Thank you for your thoughtful and detailed feedback. We deeply appreciate the time you have taken to provide a thorough analysis of our manuscript. We understand your concerns and would like to address them point by point. Below, we outline our responses and the changes we plan to implement in the revised manuscript.

Comment 1. Introduction:

  • The introduction provides an adequate historical overview of fibromyalgia; however, it would benefit from a more structured presentation of the rationale behind the study.
  • Some references appear outdated; consider including recent studies to reflect current findings.
  • The introduction should more clearly outline the study's objectives and how they address gaps in the existing literature.
  • Some sentences are lengthy and difficult to follow. In Lines 38-46, "Fibromyalgia is a complicated syndrome today due to the presence of characteristic symptoms: chronic pain, joint stiffness, fatigue, sleep problems, brain dysfunction and depression." The sentence structure can be simplified for readability. I suggested being revised to: "Fibromyalgia is a complex syndrome characterized by chronic pain, joint stiffness, fatigue, sleep disturbances, brain dysfunction, and depression."
  • Also, in Lines 58-67, there is an awkward phrasing: "It is considered that among the most important factors in triggering FM would be the reflex phenomena..." Suggested revision: "Reflex phenomena are considered among the most important factors triggering FM..."

Response 1. We appreciate your feedback regarding the Introduction. We have reorganized and reduced the information to make it clearer.

  • We have entered 6 current references in the introduction part.
  • We reorganized the information for clarity and introduced some information from the specialized literature.

Studies on mind-body recovery are limited, and further research is needed. Published studies have limited effects due to low-quality evidence or lack of recent studies [19]. At the same time, mind-body therapies are favorable in pathologies that have affected psychological function [20], aiming to induce the body's relaxation response [21]. Au-thor Liraz Cohen-Biton concludes that holistic therapies focusing on body, mind, and soul provide an appropriate response that promotes the health status of FM patients [22].          

  • Thank you. We reformulated the two sentences as you suggested.
  • Fibromyalgia is a complex syndrome characterized by chronic pain, joint stiffness, fatigue, sleep disturbances, brain dysfunction, and depression.
  • Reflex phenomena are considered among the most important factors triggering FM, that are associated with deep chronic pain produced by the activation of a pain-spasm cycle, by the presence of repeated trauma or postural stress [9].

Comment 2 Research Design

  • The study design is generally appropriate for the research objectives. However, there are concerns about the short intervention period (2 weeks), which may not be sufficient to observe meaningful changes.
  • The randomization process should be described in more detail to ensure transparency and reproducibility.
  • The inclusion of only two groups may limit the generalizability; consider discussing potential confounders in more depth.
  • There also some sentences are lengthy in Lines 165-174, "They had a complex recovery program consisting of 20-minute individual physical therapy that included aerobic exercise, stretching and muscle-strengthening exercises, 15-minute low-frequency electrotherapy, massage, and hydrokinesitherapy that facilitates movement." Please consider this revision, "Their recovery program included 20-minute sessions of physical therapy, comprising aerobic exercise, stretching, and muscle-strengthening exercises. Additionally, they received 15-minute sessions of low-frequency electrotherapy, massage, and hydrokinesitherapy."

Response 2 Thank you for your observations. All changes have been made in the text.

About the short intervention: Romanian legislation only offers 10 free recovery sessions during 2 weeks. In order to avoid differences between the two groups, we introduced the same period for the relaxation intervention.

Allocation to the study was done after the patients' visit to the rheumatologist. Following visits to the rheumatologist, 67 patients with fibromyalgia were invited to participate in the present study, by the physical therapist. 7 of these withdrew before the start of the initial assessments at the time of study presentation. A total of 60 patients participated in the study. Patients were chosen following the inclusion and exclusion criteria. The main inclusion criteria were the age of at least 18 years, the presence of a diagnosis of fibromyalgia, voluntary and independent participation in the study and the absence of contraindications for the intervention. At the same time, pregnant people and those who were absent from the evaluation were excluded from the study. Allocation to the study was done randomly, without knowing the severity of the symptoms, so that at the end of the study, the chance of specific recovery would be given to the group with more effective intervention. 30 patients out of the 60 were randomly assigned to experimental group 1 and underwent treatment at the Recovery Clinical Hospital in Băile Felix. The other 30 patients were assigned to experimental group 2 and performed the treatment at home. The period of the study is January-November 2024. All 60 patients also administered the pharmacological treatment received during the study period upon learning the diagnosis.

The allocation of patients was done by the method of simple randomization.

About of the inclusion of only two groups. Considering the low incidence of FM, the study was built on the basis of two study groups to follow two different types of intervention. At the same time, in order to be able to interpret the data from a statistical point of view, it was necessary for a group to consist of at least 30 patients.

Thank You. We reformulated the phrase about the recovery program of experimental group 1.

They had a complex recovery program consisting of 30 minutes of individual physical therapy that included warm-up and exercise preparation exercises, mobilization exercises for all joints, followed by muscle strengthening exercises and stretching at the end for increased flexibility [41- 47]. Each exercise was performed in 4 sets of 10 repetitions. The exercises were performed from dorsal decubitus, ventral decubitus, standing and quadrupedal position and were performed in breathing rhythm. Patients also had electrotherapy sessions at the BTL 5000 machine with low frequency currents TENS (transcutaneous electrical nerve stimulation) for 15 minutes to reduce chronic pain, classic swedish massage 10 minutes. They also had  group hydrokinetotherapy for 20 minutes from orthostatism at a water temperature of 36-37 degrees Celsius, which facilitates movement and reduces gravitational forces [48], the benefits of thermal water being known since ancient times [49].

Comment 3. Methods

The methods section is well-structured but lacks detail in the intervention protocol for intrinsic relaxation at home. Providing more specifics on adherence monitoring would be beneficial. The statistical methods are generally appropriate; however, clarifications on how missing data were handled should be provided. A power analysis to determine the sample size adequacy should be included.

Response 3 Thank you for your observations. All changes have been made in the text.

Experimental group 2 (EG2) performed at home, for 2 weeks without the physical supervision of a physiotherapist, 20 minutes of intrinsic relaxation specific to the author Parow, every evening before going to bed. Parow's relaxation technique is based on intrinsic relaxation, which helps to actively induce relaxation and ensures mutual inhibition between muscles and psyche. It is performed without other auxiliary tools, and can be performed at home. The author recommends maintaining the supine position in bed for 20 minutes, breathing freely. During this time, it is recommended to inhale through the nose and exhale with a long "sh" to automatically induce general muscle relaxation. All patients in this group were helped before the start of the 10 recovery sessions to perform the breathing technique correctly and communicated daily, online, with the physiotherapist to record the session performed and to offer them help if they needed it. This intervention focused only on intrinsic relaxation is based on the need to integrate a recuperative program as simplified as possible and accessible to anyone, the specialists in the field claiming that notable improvements in the clinical profile occur in people who have a high adherence to the treatment.

Thank you. Allocation of patients was done by the method of simple randomization..           

The general formula for calculating the size of the patient sample used in the study was the following:

n=(Zα/2+Zβ)2⋅(2σ2)/d2,

where:

  • n is the sample size.
  • Zα/2 = 1.96, is the critical value of the standard normal distribution (Z) corresponding to the significance level α (0.05).
  • Zβ = 0.80, is the critical value of the standard normal distribution corresponding to the power of the test (1 - β),
  • σ is the population standard deviation.
  • d is the minimum effect size we want to detect

Comment 4 Results

  • The results section provides statistical comparisons but lacks visualization (e.g., tables and figures) to support the textual descriptions.
  • Some p-values (e.g., p=0.083) are not significant, but their interpretation is unclear. Consider emphasizing clinically meaningful changes rather than purely statistical significance.
  • A discussion of within-group variations should be expanded.
  • In Lines 328-334, this sentence is complex: "In the experimental group 1 that performed hospital recovery (EG1), the results show that the severity of fatigue (FSS) was significantly reduced, which suggests a general improvement in the state of fatigue." Please consider revising it. 

Response 4 Thank you for your comments, all corrections in the text have been made.

We added a figure to see the effects of the intervention in the two groups.

About the data interpretation. We have restructured this information from a clinical point of view. In table 7 and figure 1, the intervention applied to EG1 shows that sleep duration increased, with patients able to fall asleep faster and maintain sleep for a longer peri-od. Diurnal dysfunction caused by lack of sleep was reduced, with patients reporting better overall sleep quality with a moderate effect size of 0.55 points. Fatigue Severity (FSS) was significantly reduced, p=0.00, with a moderate effect size of 0.77 points, suggesting an overall improvement in fatigue status. Overall, the intervention had a significant positive effect on various aspects of sleep and daytime functioning.

In EG2, the intervention had limited effects. No significant changes were observed between baseline and final values in most sleep components, such as sleep quality, sleep duration, sleep efficiency, sleep medication use, and fatigue severity scale. The global sleep quality score (PSQI) decreased slightly, with a low effect size of 0.09 points, and the fatigue state remained at the same values.

About the discussion of within-group variations. We added a paragraph to the discussion section.

Regarding within-group variations, at EG1, the decrease in mean score indicates an improvement in patient-reported fatigue between the two assessments, with a general trend observed and improvement in fatigue symptoms for most participants, not just those with scores extremes. The standard deviation decreased from 10.811 to 8.763, suggesting a leveling of fatigue severity across participants. In the beginning, the variability of the scores was greater, but at the end, the differences between the participants became smaller, which may indicate a generalized and more even improvement. The decrease in values for each component and the total PSQI suggests an overall improvement in sleep quality and a reduction in sleep-related problems following the intervention.

Intra-group variations within EG2 show that there is no change in the mean score between the initial and final assessment, suggesting that the intervention did not influence the participants' perceived level of fatigue. Both the mean and median constants indicate that the fatigue felt by the group remained uniform before and after. The standard deviation of 12.820 indicates moderate variability between participants, both at baseline and at baseline.

We reformulated the entire paragraph. In Table 7 and Figure 1, the intervention applied to EG1 shows that sleep duration increased, with patients able to fall asleep faster and maintain sleep for a longer period. Diurnal dysfunction caused by lack of sleep was reduced, with patients reporting better overall sleep quality with a moderate effect size of 0.55 points. Fatigue Severity (FSS) was significantly reduced, p=0.00, with a moderate effect size of 0.77 points, suggesting an overall improvement in fatigue status. Overall, the intervention had a significant positive effect on various aspects of sleep and daytime functioning.

Comment 5. Conclusions

  • The conclusions align with the study findings, but they should better acknowledge the limitations regarding the short intervention period and sample size.
  • Recommendations for future research should be expanded to include longer follow-up periods and additional outcome measures.

Overall, the manuscript requires language improvements, including grammatical corrections, simplification of complex sentences, and improved clarity.

Response 5. Thank you for your comments, all corrections in the text have been made.

            About the conclusions. Due to the unclear pathogenesis of fibromyalgia, the multidisciplinary approach focused on improving flexibility, extrinsic relaxation and muscle tone has a better effectiveness on sleep quality than therapy focused only on intrinsic relaxation. Although the intervention was of short duration, the daily sessions had an increased contribution to the recovery.

About the follow-up evaluation. We decided to follow up for 6 months to follow the effects of the relaxation and to introduce new measures depending on the evaluation results. To observe the long-term results, the patients in experimental group 1 will be re-evaluated 6 months after the intervention, and the patients in experimental group 2 received the recommendation to continue the relaxation technique at home for 6 months, monitored online by a physiotherapist to avoid biases. The purpose of continuing the program is to observe if a treatment based on a single form of relaxation can improve the state of fatigue, by acting on sleep, following this period to begin multidisciplinary treatment if no beneficial results appear on the variables evaluated.

Comments on the Quality of English Language

Some sentences are lengthy and difficult to follow. Revising for clarity and conciseness is recommended. Minor grammatical errors and awkward phrasing should be corrected to enhance readability.

Response. Thank you for your comments, all corrections in the text have been made.

We believe these revisions will address your concerns and enhance the clarity and scientific rigor of the manuscript. Thank you again for your valuable input, and we look forward to submitting the revised manuscript.

Sincerely,

Authors

Reviewer 2 Report

Comments and Suggestions for Authors

This study is a comparative cohort study on sleep quality and fatigue of two recovery interventions (standard multidisciplinary intervention in a recovery hospital and intrinsic relaxation therapy at home) in patients with fibromyalgia. It will be of interest to many readers. Please consider the following:

1. Please provide references for all sentences presented in the introduction, if possible.

Fibromyalgia is a complicated syndrome today due to the presence of characteristic symptoms: chronic pain, joint stiffness, fatigue, sleep problems, brain dysfunction and depression (References).

2. The content on sleep in the third paragraph of the introduction is too long and difficult to read. Please summarize the content on sleep.

3. The advantages of Standard Care versus Intrinsic Relaxation at Home on Physiological Parameters or previous studies should be mentioned separately in the introduction.

4. There should be an effect size for the number of subjects. If possible, please add it.

5. Please also describe the method of assigning subjects. 6. Aerobic exercise [37], stretching [38] and muscle-strengthening exercises [39], 15-minute low-frequency electrotherapy, massage, and hydrokinesitherapy that facilitates movement [40].

-> Please provide detailed information on the type of exercise and the number of times, and information on the electrotherapy equipment is also needed. Massage, and hydrokinesitherapy also require detailed information. They are important parts of the research method.

7. Although abbreviations are used repeatedly, please use them in accordance with the format.

8. Why did you use Std. Error Mean in Table 1?

9. It is difficult to see the difference between groups. Please indicate the difference between groups so that we can see it.

10. There are too many paragraphs in the discussion paragraph. Please merge paragraphs with the same content, and explain the difference between groups.

Author Response

Dear Reviewer,

Thank you for your thoughtful and detailed feedback. We deeply appreciate the time you have taken to provide a thorough analysis of our manuscript. We understand your concerns and would like to address them point by point. Below, we outline our responses and the changes we plan to implement in the revised manuscript.

Comment  

This study is a comparative cohort study on sleep quality and fatigue of two recovery interventions (standard multidisciplinary intervention in a recovery hospital and intrinsic relaxation therapy at home) in patients with fibromyalgia. It will be of interest to many readers. Please consider the following:

  1. Please provide references for all sentences presented in the introduction, if possible.

Fibromyalgia is a complicated syndrome today due to the presence of characteristic symptoms: chronic pain, joint stiffness, fatigue, sleep problems, brain dysfunction and depression (References).

  1. The content on sleep in the third paragraph of the introduction is too long and difficult to read. Please summarize the content on sleep.
  2. The advantages of Standard Care versus Intrinsic Relaxation at Home on Physiological Parameters or previous studies should be mentioned separately in the introduction.
  3. There should be an effect size for the number of subjects. If possible, please add it.
  4. Please also describe the method of assigning subjects.
  5. Aerobic exercise [37], stretching [38] and muscle-strengthening exercises [39], 15-minute low-frequency electrotherapy, massage, and hydrokinesitherapy that facilitates movement [40].

- Please provide detailed information on the type of exercise and the number of times, and information on the electrotherapy equipment is also needed. Massage, and hydrokinesitherapy also require detailed information. They are important parts of the research method.

  1. Although abbreviations are used repeatedly, please use them in accordance with the format.
  2. Why did you use Std. Error Mean in Table 1?
  3. It is difficult to see the difference between groups. Please indicate the difference between groups so that we can see it.
  4. There are too many paragraphs in the discussion paragraph. Please merge paragraphs with the same content, and explain the difference between groups.

Response 1. Thank you for your comments, all corrections in the text have been made.

  1. We have inserted references to each paragraph. Fibromyalgia is a complex syndrome characterized by chronic pain, joint stiffness, fatigue, sleep disturbances, brain dysfunction, and depression [1-2].
  2. We summarized the paragraph about sleep and made it clearer. Regarding sleep, it is known that there are two types of sleep: REM and non-REM. The non-REM period has 4 stages whose waves have certain electroencephalographic characteristics [29-30]. In fibromyalgia, there are disturbances in the fourth phase of non-REM sleep, which are translated by the appearance of alpha waves that overlap the normal, delta waves. The results from specialized literature show that sleep disorders are one of the most important aspects of FM [31] poor sleep quality being a fundamental index in the pathophysiology of FM, together with fatigue and pain. The prevalence of sleep quality shows that only 11% of FM patients reported good sleep quality, while up to 99% of patients report having problems with the amount, initia-tion, maintenance of sleep during the night, with long refresh times or they wake up too early in the morning, which makes it difficult to start their daily activities [32]. Kline et al[33] mentioned that the definition of sleep quality should be related to the per-sonal experience of each person, depending on the stage that affects them. Poor sleep quality has a negative impact, because it lowers the pain threshold and increases the level of its catastrophizing. At the same time, this cycle of sleep impairment causes a high level of fatigue, studies show that poor sleep quality leads to the appearance of cognitive, social and integration problems [34], having a tendency to become chronic [35]. Patients with fibromyalgia complain of sleep disturbances and the intensification of pain, stiffness and fatigue after waking up [36]. Often, when patients are asked if they wake up rested, the answer is negative, with sleep being impaired in over 80% of FM patients [37]. Often sleepless nights are followed by worsening pains [38]. Some studies suggest that fatigue is a cause of FM and not a symptom [39]. In the anamnesis, there are frequent associations with diseases that recognize a stress etiology, such as ulcerative hemorrhagic colitis, irritable colon, migraine, hypertension, so on. [40].
  3. We reorganized the information about physical exercises in paragraph 2 and about relaxation in paragraph 3.
  4. The general formula for calculating the size of the patient sample used in the study was the following:

n=(Zα/2+Zβ)2⋅(2σ2)/d2,

where:

  • n is the sample size.
  • Zα/2 = 1.96, is the critical value of the standard normal distribution (Z) corresponding to the significance level α (0.05).
  • Zβ = 0.80, is the critical value of the standard normal distribution corresponding to the power of the test (1 - β),
  • σ is the population standard deviation.
  • d is the minimum effect size we want to detect

  1. About the method of assigning subjects. Following visits to the rheumatologist, 67 patients with fibromyalgia were invited to participate in the present study, by the physical therapist. 7 of these withdrew before the start of the initial assessments at the time of study presentation. A total of 60 patients participated in the study. Patients were chosen following the inclusion and exclusion criteria. The main inclusion criteria were the age of at least 18 years, the presence of a diagnosis of fibromyalgia, voluntary and independent participation in the study and the absence of contraindications for the intervention. At the same time, pregnant people and those who were absent from the evaluation were excluded from the study. Allocation of patients was done by the method of simple randomization without knowing the severity of the symptoms, so that at the end of the study, the chance of specific recovery would be given to the group with more effective intervention. 30 patients out of the 60 were randomly assigned to experimental group 1 and underwent treatment at the Recovery Clinical Hospital in Băile Felix. The other 30 patients were assigned to experimental group 2 and performed the treatment at home. The period of the study is January-November 2024. All 60 patients also administered the pharmacological treatment received during the study period upon learning the diagnosis.
  2. We added information about the EG1 program. EG1 performed 10 standard recovery sessions in the presence of a physical therapist. They had a complex recovery program consisting of 30 minutes of individual physical therapy that included warm-up and exercise preparation exercises, mobilization exercises for all joints, followed by muscle strengthening exercises and stretching at the end for increased flexibility [41- 47]. Each exercise was performed in 4 sets of 10 repetitions. The exercises were performed from dorsal decubitus, ventral decubitus, standing, and quadrupedal position and were performed in breathing rhythm. Patients also had electrotherapy sessions at the BTL 5000 machine with low-frequency currents TENS (transcutaneous electrical nerve stimulation) for 15 minutes to reduce chronic pain, and classic Swedish massage 10 minutes. They also had group hydrokinetotherapy for 20 minutes from orthostatism at a water temperature of 36-37 degrees Celsius, which facilitates movement and reduces gravitational forces [48], the benefits of thermal water being known since ancient times [49]. Exercise is part of the management of fibromyalgia, as it improves pain, sleep, and general functioning [45].
  3. About the abbreviations. Thank you. We have revised all abbreviations to avoid mistakes.
  4. We redid Table 1 and excluded the standard error. Thank you.
  5. About the difference between groups. We added a figure to clearly illustrate the effects of the intervention in each group and added a paragraph to discuss differences between groups. Regarding within-group variations, at EG1, the decrease in mean score indicates an improvement in patient-reported fatigue between the two assessments, with a general trend observed and improvement in fatigue symptoms for most participants, not just those with scores extremes. The standard deviation decreased from 10.811 to 8.763, suggesting a leveling of fatigue severity across participants. At the beginning, the variability of the scores was greater, but at the end, the differences between the participants became smaller, which may indicate a generalized and more even improvement. The decrease in values for each component and the total PSQI suggests an overall improvement in sleep quality and a reduction in sleep-related problems following the intervention. Intra-group variations within EG2 show that there is no change in the mean score between the initial and final assessment, suggesting that the intervention did not influence the participants' perceived level of fatigue. Both the mean and median constants indicate that the fatigue felt by the group remained uniform before and after. The standard deviation of 12.820 indicates moderate variability between participants, both at baseline and at baseline.
  6. Thank you. We restructured the discussion chapter and added information about differences between groups.

We believe these revisions will address your concerns and enhance the clarity and scientific rigor of the manuscript. Thank you again for your valuable input, and we look forward to submitting the revised manuscript.

Sincerely,

Authors

Reviewer 3 Report

Comments and Suggestions for Authors

Dear authors,
Thank you for the opportunity to review your interesting manuscript. First of all, I would like to congratulate all of you on this work, which effectively summarizes important and previous research.
I will give my feedback following the structure of the manuscript.
1.Title and abstract The title is informative and the abstract provides a summary of the manuscript's major aspects. No further comments.
2.Introduction The background chapter is clear, well-developed, and thoroughly referenced, with a strong and logical structure. However, in my opinion, it is overly lengthy. The authors might consider making an effort to streamline the first part of this section, as it primarily serves to contextualize FM and could be more concise .
3.Materials and Methods First of all, I would like to congratulate the authors on this section, which I believe is very well-written and provides a great deal of valuable information. However, I would like to offer a few comments that might help improve it further. -I think it would be interesting to clarify whether any type of sampling was conducted and who invited the participants to take part in the study. -The information provided between lines 198 and 203 seems more appropriate for the limitations section, which should ideally be placed immediately after or at the end of the discussion. -The ethical aspects mentioned in this section might benefit from being presented in a dedicated section on ethical considerations. Instruments In my opinion, this section is correct, but it would be more effective to present the information on data analysis in a separate section for better clarity. Additionally, it would be helpful to include the bibliographic references for all the instruments used, such as the FSS-9 and PSQI, to ensure proper citation. Assessment of fatigability, fatigue and Assessment of overall sleep quality This section is very complete and well-written, with nothing to add.
4.Results I would like to congratulate the authors on this section, which in my opinion is very well done. However, I believe that the results presented under the name 'patient data' are part of the results, and in my opinion, the authors would include them in this section. I would also recommend that the tables be cited within the text. Currently, Table 1 and Table 2 are not referenced in any of the paragraphs.

Author Response

Dear Reviewer,

Thank you for your thoughtful and detailed feedback. We deeply appreciate the time you have taken to provide a thorough analysis of our manuscript. We understand your concerns and would like to address them point by point. Below, we outline our responses and the changes we plan to implement in the revised manuscript.

  1. Title and abstract. The title is informative and the abstract provides a summary of the manuscript's major aspects. No further comments.

Comment 1 Introduction. The background chapter is clear, well-developed, and thoroughly referenced, with a strong and logical structure. However, in my opinion, it is overly lengthy. The authors might consider making an effort to streamline the first part of this section, as it primarily serves to contextualize FM and could be more concise.

Response 1. Thank you for your comments, all corrections in the text have been made.

We appreciate your feedback regarding the Introduction. We have reduced the information and we reorganized it.

Comment 2. Materials and Methods First of all, I would like to congratulate the authors on this section, which I believe is very well-written and provides a great deal of valuable information. However, I would like to offer a few comments that might help improve it further.

-I think it would be interesting to clarify whether any type of sampling was conducted and who invited the participants to take part in the study.

Response 2. Thank you for your comments, all corrections in the text have been made. The allocation of patients was done by the method of simple randomization.

Following visits to the rheumatologist, 67 patients with fibromyalgia were invited to participate in the present study, by the physical therapist. 7 of these withdrew before the start of the initial assessments at the time of study presentation. A total of 60 patients participated in the study. Allocation to the study was done randomly, without knowing the severity of the symptoms, so that at the end of the study, the chance of specific recovery would be given to the group with more effective intervention.

-The information provided between lines 198 and 203 seems more appropriate for the limitations section, which should ideally be placed immediately after or at the end of the discussion.

Thank you for your comments. We moved this paragraph to the end of the discussions.

-The ethical aspects mentioned in this section might benefit from being presented in a dedicated section on ethical considerations. Instruments In my opinion, this section is correct, but it would be more effective to present the information on data analysis in a separate section for better clarity.

Thank you for your comments. We moved the information about ethics to the dedicated section and moved the data about statistics to subsection 2.2.

- Additionally, it would be helpful to include the bibliographic references for all the instruments used, such as the FSS-9 and PSQI, to ensure proper citation. Assessment of fatigability, fatigue and Assessment of overall sleep quality. This section is very complete and well-written, with nothing to add.

Thank You. References 51 and 52 contain information about the questionnaires used.

Comment 3 Results

I would like to congratulate the authors on this section, which in my opinion is very well done.

However, I believe that the results presented under the name,patient data,; are part of the results, and in my opinion, the authors would include them in this section. I would also recommend that the tables be cited within the text. Currently, Table 1 and Table 2 are not referenced in any of the paragraphs.

Response 3. Thank you for your comments, all corrections in the text have been made.

We moved the section about patients to the results and highlighted tables 1 and 2 in the text.

We believe these revisions will address your concerns and enhance the clarity and scientific rigor of the manuscript. Thank you again for your valuable input, and we look forward to submitting the revised manuscript.

Sincerely,

Authors

Round 2

Reviewer 1 Report

Comments and Suggestions for Authors

The authors have made noticeable efforts to improve the clarity and depth of their study. The introduction has been refined, and additional references have been incorporated to provide a more comprehensive background on fibromyalgia and its impact on sleep quality and fatigue. These additions enhance the study's relevance and provide better context for the research questions.

The methodology section is now more structured, with clearer descriptions of participant selection, randomization, and the interventions applied. The authors have specified the inclusion/exclusion criteria more explicitly and have justified their study design, which improves the overall rigor of the study.

Comments on the Quality of English Language

The manuscript still contains some minor grammatical errors and awkward phrasing. While the readability has improved, additional proofreading would further refine the clarity and coherence of the writing.

Reviewer 2 Report

Comments and Suggestions for Authors

All properly reflected. Thank you for your hard work in research.